# Maternal smoking during pregnancy and offspring body composition in adulthood: Results from two birth cohort studies

Elma Izze da Silva Magalhães, Natália Peixoto Lima, Ana Maria Baptista Menezes, Helen Gonçalves, Fernando C Wehrmeister, Maria Cecília Formoso Assunção, Bernardo Lessa Horta

Postgraduate Program in Epidemiology, Federal University of Pelotas, Pelotas, Rio Grande do Sul, Brazil

**Correspondence to**
Elma Izze da Silva Magalhães; elmaizzenutri@gmail.com

## ABSTRACT

**Objective** To evaluate the association of maternal smoking during pregnancy with offspring body composition in adulthood and explore the causality of this association.

**Design** Birth cohort.

**Setting** Population-based study in Pelotas, Brazil.

**Participants** All newborn infants in the city's hospitals were enrolled in 1982 and 1993. At a mean age of 30.2 and 22.6 years, the 1982 and 1993 cohorts, respectively, followed the subjects and 7222 subjects were evaluated.

**Primary outcome measures** Body mass index (BMI), fat mass index, android to gynoid fat ratio, waist circumference, waist to height ratio, lean mass index and height.

**Results** Prevalence of maternal smoking during pregnancy was 35.1% and 32.6%, in 1982 and 1993 cohorts, respectively. Offspring of smoking mothers showed higher mean BMI (β: 0.84; 95% CI: 0.55 to 1.12 kg/m$^2$), fat mass index (β: 0.44; 95% CI: 0.23 to 0.64 kg/m$^2$), android to gynoid fat ratio (β: 0.016; 95% CI: 0.010 to 0.023), waist circumference (β: 1.74; 95% CI: 1.15 to 2.33 cm), waist to height ratio (β: 0.013; 95% CI: 0.010 to 0.017) and lean mass index (β: 0.33; 95% CI: 0.24 to 0.42 kg/m$^2$), whereas height was lower (β: −0.95; −1.26 to −0.65). Weight gain in the first 2 years captured most of the association of maternal smoking with BMI (96.2%), waist circumference (86.1%) and fat mass index (71.7%).

**Conclusions** Maternal smoking in pregnancy was associated with offspring body composition measures in adulthood.

## Strengths and limitations of this study

► High follow-up rate and the lack of differences in the follow-up rate according to maternal smoking during pregnancy, suggesting that selection bias is unlikely.

► We increased causal inference by comparing the estimates of maternal and paternal smoking.

► Body composition was assessed using dual X-ray absorptiometry and air-displacement plethysmography.

► We evaluated smoking in pregnancy using a questionnaire and the information was not validated with biochemical markers.

► Information on partner smoking was obtained at different times in the two cohorts.

## INTRODUCTION

Maternal smoking during pregnancy has clear short-term consequences, increasing the risk of low birth weight and intrauterine growth retardation.[1] Moreover, maternal smoking is also associated with increased infant mortality rate, and cessation of smoking during pregnancy reduces the risk of infant death.[2] However, evidence on the long-term consequences of maternal smoking are not clear-cut. Meta-analysis by Brion *et al*[3] related that blood pressure was slightly higher among offspring of smoking mothers. It has also been reported a higher risk of diabetes.[4] On the other hand, Horta *et al*[5] observed that the association between maternal smoking in the pregnancy and metabolic cardiovascular risk factors in early adulthood was mediated by offspring lifestyle in adulthood.

With respect to body composition, it has been suggested that maternal smoking in pregnancy is positively associated with adiposity and risk of overweight/obesity in the offspring.[6 7] On the other hand, because socioeconomic status is negatively associated with maternal smoking[8] and obesity,[9] the association of maternal smoking with offspring obesity could be due to residual confounding.[10] Indeed, some studies have reported that after controlling for socioeconomic and environmental variables, the magnitude of the association decreased.[11 12]

Because socioeconomic status is usually assessed using few variables, the full

dimension of socioeconomic status may not have been captured. And, residual confounding may be observed even among those studies that adjusted the estimates to socioeconomic and demographic variables. Comparison of the magnitude of the associations with maternal and paternal smoking is a strategy that can be used to increase causal inference. Similar associations for paternal and maternal smoking would suggest that confounding by socioeconomic, environmental or familiar variables is an explanation for the observed associations. On the other hand, if the association is stronger for maternal variables, it is unlikely that the observed association is due to residual confounding.[13]

This study was aimed at assessing the association of maternal smoking during pregnancy with offspring body composition in adulthood and explore the causality of this association.

## METHODS

### Study design and population

This study is based on data from two birth cohorts carried out in Pelotas, a southern Brazilian city. In 1982 and 1993, in the perinatal study, all maternity hospitals in the city were visited daily, and the births identified. Those liveborns whose families lived in the urban area of the city were examined and their mothers interviewed soon after delivery. These subjects have been followed-up for several times through the life cycle, and further details on the study methodology have been published elsewhere.[14 15]

Briefly, during the whole of 1982, among the births to mothers living in the urban area of Pelotas, 5914 newborns and their mothers were enrolled and accepted to participate in the perinatal study. Then, follow-up visits were carried at the ages of 1, 2, 4, 13, 15, 18, 19, 23 and 30 years. Most visits included subsamples of the cohort, except for those at 2, 4, 23 and 30 years, in which we attempted to locate the whole cohort. Between June 2012 and February 2013, 4534 members were located, in which 467 were living far from Pelotas, 86 refused and another 280 did not attend the interview in spite of repeated invitations. Thus, a total of 3701 subjects were interviewed at age 30 years, which adding to the number of participants known to have died (n=325) made up 68.1% of the original cohort.

Similarly, among the births occurring in Pelotas during the year 1993 (n=5265), 5249 subjects and their mothers were enrolled and agreed to take part in the longitudinal study. Subsamples of the cohort were evaluated in the follow-up visits and were carried at the ages of 1, 3 and 6 months and at 1 and 4 years. In the follow-up visits at 11, 15, 18 and 22 years, the whole cohort was recruited. The follow-up at 22 years occurred from October 2015 to July 2016, and of the original cohort, 4933 were found. Among the located members, 175 refused to participate and 1071 were considered losses. Those who were interviewed (n=3810), added to those known to have died (n=193), represent a retention rate of 76.3%. The follow-up rates

at each follow-up visit of the 1982 and 1993 cohort are described in online supplementary table 1.

### Exposure variables

In both cohorts, information on maternal smoking during pregnancy, duration and numbers of cigarettes smoked per day were obtained retrospectively from the mothers shortly after delivery, in the perinatal study. Those mothers who reported any tobacco use during pregnancy were considered as smokers, and they were asked whether they had smoked during the whole pregnancy and how many cigarettes on average they had smoked per day. In addition, we also collected information on partner smoking. In the 1993 cohort, this information was gathered based on the mothers report, in the perinatal study, whereas in the 1982 cohort, information on partner smoking was provided by the mother in the follow-up visit at 4 years of age.

### Outcome variables

In follow-up at 30 and 22 years, subjects were invited to visit the research clinic to be interviewed, examined and donate a blood sample. Weight was measured using the Bod Pod scale and height with a portable stadiometer (accuracy of 0.1 cm). Body mass index (BMI) was estimated by dividing weight (in kg) by square height (in metres).

Waist circumference (in cm) was measured with the subject standing, with the arms hanging freely and next to the body, using a non-elastic measuring tape in the horizontal plane around the narrowest part of the waist. In obese subjects, the measure was taken in the horizontal plane at the point between the last rib and the iliac crest. Waist to height ratio was estimated by dividing waist circumference by height (both in cm).

Fat-free and fat mass (total and in the android and gynoid regions) were assessed using dual-energy X-ray absorptiometry (DXA). Lean and fat mass indexes were estimated by dividing fat-free and fat mass (in kg) by square height in metres. Android to gynoid fat ratio was estimated by dividing percentage fat in region android by percentage fat in gynoid region. Pregnant women or those in the postpartum period (3 months) were excluded from this assessment.

### Confounders

The following variables were considered as possible confounders: sex, age in the assessment of the outcomes, family income at birth (total income in the month before the interview), maternal skin colour (self-reported), maternal schooling (years of schooling successfully completed), maternal age at childbirth, parity (number of previous deliveries) and pre-gestational nutritional status. Information about sex of the child, family income at birth, maternal skin colour, maternal schooling, maternal age at childbirth and parity were collected through a questionnaire applied to the mothers in the interview of the perinatal study. To evaluate the pre-gestational nutritional

status, information on prepregnancy maternal weight was obtained from the antenatal care records or by maternal recall and height was measured by the research team in the interview immediately after delivery. The exact age in the assessment of outcomes was reported by subjects in the 30 and 22 years follow-up in the 1982 and 1993 cohorts, respectively.

## Mediators

Birth weight and weight gain from birth to 2 years were considered as possible mediators. Birth weight (in grams) was assessed soon after delivery, by the hospital staff, using paediatric scales that were calibrated weekly by the research team. Weight gain from birth to 2 years of age was obtained by the difference in z-score between the child's weight at 2 years of age and the birth weight. The weight of the child at 2 years of age was measured only in the follow-up of the 1982 cohort. Thus, this analysis was only performed only for the 1982 cohort.

## Statistical analysis

Stata, V.14.0 (StataCorp, College Station, TX, USA) was used in the analysis. Data from the two cohorts were analysed together because both cohorts showed similar association between maternal smoking and offspring anthropometry (data not shown). Means were compared using Student's t-test and analysis of variance. Multiple linear regression was used to obtain estimates that were adjusted for confounding variables. G-computation (Stata package: st0238) was used to estimate direct and indirect effects of maternal smoking in pregnancy on offspring body composition in adulthood (bootstrap replications: 10,000). In the analysis for birth weight, gestational age was considered as post-confounder (confounders in the relationship between the mediator and the outcome), while for weight gain from birth to 2 years, we considered family income in childhood as a post-confounder. Sex, age at body composition assessment, family income at birth, parity, pre-gestational BMI, skin colour, maternal schooling and height were included as base-confounders.

## Patient and public involvement

The study was approved by the Ethical Review Board of the Faculty of Medicine of the Federal University of Pelotas and was performed in accordance with the ethical standards established in the 1964 Declaration of Helsinki and its later amendments. Written informed consent was obtained from participating subjects.

## RESULTS

Among the 3701 and 3810 participants of the 1982 and 1993 cohorts on follow-up at 30 and 22 years, respectively, information on maternal smoking and at least one of the outcomes were available for 3626 participants from 1982 cohort and 3596 from 1993. Analysing the follow-up rates in the last visit of each cohort, according to baseline characteristics, was observed that in the 1982 the cohort

follow-up rates at 30 years were slightly higher among females, those who were born preterm and those in the intermediate socioeconomic categories. In the 1993 cohort, losses to follow-up were higher among males and at the extremes of the income distribution. Nevertheless, follow-up rates among different subgroups are reasonably similar, ranging from 60% to 75% in all variables studied so that attrition bias is unlikely (online supplementary table 2).

Table 1 shows the characteristics of the subjects included in the analyses. The proportion of preterm births was 5.6% and 8.0% in 1982 and 1993, respectively. The prevalence of overweight/obesity was 43.2% at 22 years (1993 cohort) and 57.6% at 30 years (1982 cohort). Prevalence maternal smoking during pregnancy was 35.0% and 32.4% in 1982 and 1993, respectively. In the 1993 cohort, 32.4% of the partners smoked during pregnancy, whereas, in the 1982 cohort, 58.8% of the partners were smoker at the follow-up visit at 4 years of age.

Online supplementary table 3 shows that prevalence of maternal smoking during whole pregnancy was significantly higher among the mothers in the lower income tertile and lower schooling in both cohorts. On the other hand, maternal smoking during pregnancy was associated only with the partner smoking in pregnancy, in the 1993 cohort, but no with the partner smoking at 4 years, in the 1982 cohort.

In the crude analyses (table 2), maternal smoking in the pregnancy was associated with higher mean BMI, fat mass index, android to gynoid fat ratio, waist circumference, waist to height ratio, as well as lean mass index (p<0.05). On the other hand, offspring of mothers who smoked during pregnancy had lower height (p<0.0001). With respect to duration of maternal smoking during pregnancy, body mass, fat mass index, android to gynoid fat ratio, waist circumference, waist to height ratio and lean mass index were slightly higher among the offspring of mothers who smoked only in part of pregnancy in relation to those whose mother smoked in the whole pregnancy. But the lower means were observed among those subjects whose mothers did not smoke in the pregnancy (p<0.05). On the other hand, height was inversely associated with the duration of maternal smoking during pregnancy (p<0.0001). Regarding partner, smoking, no significant differences were observed for partner smoking at 4 years (cohort 1982). On the other hand, lower means of waist circumference, waist to height ratio and lean mass index and height were observed in the offspring in which the partners smoked during pregnancy (p<0.05).

After controlling for confounding variables, offspring of smoking mothers showed higher BMI, fat mass index, android to gynoid fat ratio, waist circumference, waist to height ratio and lean mass index, whereas height was lower. Concerning the duration and intensity of maternal smoking during pregnancy, we only observed a dose–response association for height, whereas for the remaining body composition outcomes, the regression coefficients tend to slightly higher among those subjects

**Table 1** Characteristics of population included in the present study, Pelotas, Brazil

| Variable | 1982 Cohort (n=3626) | | 1993 Cohort (n=3596) | |
|---|---|---|---|---|
| | N (%)* | Mean (SD)† | N (%) * | Mean (SD)† |
| Sex | | | | |
| Male | 1766 (48.7) | | 1706 (47.4) | |
| Female | 1860 (51.3) | | 1890 (52.6) | |
| Gestational age (weeks) | | | | |
| <37 | 164 (5.6) | | 283 (8.0) | |
| 37–39 | 1336 (45.8) | | 2741 (77.2) | |
| ≥40 | 1420 (48.6) | | 527 (14.8) | |
| Birth weight (g) | | | | |
| <2500 | 280 (7.7) | | 338 (9.4) | |
| 2500–2999 | 956 (26.4) | | 942 (26.2) | |
| ≥3000 | 2389 (65.9) | | 2314 (64.4) | |
| Skin colour | | | | |
| White | 2750 (75.8) | | 2163 (63.5) | |
| Non-white | 876 (24.2) | | 1245 (36.5) | |
| Maternal smoking during pregnancy | | | | |
| Non-smokers | 2353 (65.0) | | 2430 (67.6) | |
| Smoked in part of pregnancy | 276 (7.6) | | 160 (4.4) | |
| Smoked in whole pregnancy | 990 (27.4) | | 1006 (28.0) | |
| Partner smoking‡ | | | | |
| No | 1166 (41.2) | | 2433 (67.6) | |
| Yes | 1665 (58.8) | | 1163 (32.4) | |
| Familiar income at birth (minimum wage) | | | | |
| ≤1 | 715 (19.8) | | 628 (17.8) | |
| 1.1–3.0 | 1780 (49.3) | | 1472 (41.7) | |
| 3.1–6.0 | 706 (19.6) | | 877 (24.9) | |
| >6.0 | 408 (11.3) | | 550 (15.6) | |
| Maternal schooling (years) | | | | |
| 0–4 | 2719 (75.1) | | 950 (26.5) | |
| 5–8 | 399 (11.0) | | 1695 (47.2) | |
| ≥9 | 503 (13.9) | | 946 (26.3) | |
| Body mass index in adulthood (kg/m$^2$) | | 26.8 (5.5) | | 25.2 (5.3) |
| <18.5 | 71 (2.0) | | 168 (4.7) | |
| 18.5–24.9 | 1426 (40.4) | | 1855 (52.1) | |
| 25.0–29.9 | 1224 (34.7) | | 959 (27.0) | |
| ≥30.0 | 810 (22.9) | | 577 (16.2) | |
| Fat mass index (kg/m$^2$) | | 8.7 (4.3) | | 7.9 (4.5) |
| Android to gynoid fat ratio | | 0.50 (0.14) | | 0.42 (0.12) |
| Waist circumference (cm) | | 84.8 (12.6) | | 80.0 (11.6) |
| Waist to height ratio | | 0.51 (0.07) | | 0.48 (0.07) |
| Lean mass index (kg/m$^2$) | | 16.7 (2.7) | | 16.1 (2.6) |
| Height (cm) | | 167.7 (9.2) | | 167.4 (9.5) |

*For the categorical variables.
†For the continuous variables.
‡At 4 years for the 1982 cohort and in pregnancy for the 1993 cohort.
IQI, inter quartiles interval.

**Table 2** Mean for anthropometric and body composition measures in adulthood and according to maternal smoking during pregnancy and partner smoking

| | Mean (SD) | | | | | | |
|---|---|---|---|---|---|---|---|
| | Body mass index (kg/m²) | Fat mass index (kg/m²) | Android to gynoid fat ratio | Waist circumference (cm) | Waist to height ratio | Lean mass index (kg/m²) | Height (cm) |
| Maternal smoking in pregnancy† | p<0.0001 **** | p=0.0090** | p<0.0001**** | p=0.0001**** | p=0.0001**** | p=0.0001**** | p=0.0001**** |
| No | 25.8 (5.4) | 8.2 (4.3) | 0.46 (0.13) | 82.0 (12.3) | 0.49 (0.07) | 16.3 (2.6) | 168.1 (9.4) |
| Yes | 26.4 (5.6) | 8.5 (4.6) | 0.47 (0.14) | 83.2 (12.5) | 0.50 (0.07) | 16.6 (2.6) | 166.6 (9.3) |
| Duration of maternal smoking‡ | p<0.0001**** | p=0.0036** | p<0.0001**** | p<0.0001**** | p<0.0001**** | p<0.0001**** | p<0.0001**** |
| Non-smokers | 25.8 (5.4) | 8.2 (4.3) | 0.46 (0.13) | 82.0 (12.3) | 0.49 (0.07) | 16.3 (2.6) | 168.1 (9.4) |
| Smoked in part of pregnancy | 27.0 (5.6) | 9.0 (4.6) | 0.48 (0.13) | 84.7 (12.5) | 0.50 (0.07) | 16.7 (2.6) | 167.8 (9.6) |
| Smoked in whole pregnancy | 26.3 (5.6) | 8.4 (4.6) | 0.47 (0.14) | 82.9 (12.4) | 0.50 (0.07) | 16.6 (2.6) | 166.3 (9.2) |
| Cigarettes smoked per day‡ | p=0.0001 **** | p=0.0165‡ | p<0.0001**** | p=0.0007 † | p<0.0001**** | p<0.0001**** | p<0.0001**** |
| None | 25.8 (5.4) | 8.2 (4.3) | 0.46 (0.13) | 82.0 (12.3) | 0.49 (0.07) | 16.3 (2.6) | 168.1 (9.4) |
| 1–14 cigarettes/day | 26.5 (5.6) | 8.6 (4.6) | 0.47 (0.14) | 83.2 (12.5) | 0.50 (0.07) | 16.6 (2.6) | 166.7 9.3) |
| ≥15 cigarettes/day | 26.3 (5.4) | 8.3 (4.5) | 0.48 (0.14) | 83.2 (12.5) | 0.50 (0.07) | 16.7 (2.5) | 165.9 (9.4) |
| **1982 cohort** | | | | | | | |
| Partner smoking at 4 years * | p=0.4287 | p=0.9976 | p=0.3031 | p=0.3249 | p=0.5670 | p=0.3963 | p=0.1918 |
| No | 26.8 (5.5) | 8.4 (4.4) | 0.50 (0.14) | 84.9 (12.6) | 0.51 (0.07) | 16.7 (2.6) | 167.9 (9.1) |
| Yes | 26.7 (5.4) | 8.4 (4.3) | 0.49 (0.14) | 84.4 (12.4) | 0.50 (0.07) | 16.6 (2.7) | 167.4 (9.3) |
| **1993 cohort** | | | | | | | |
| Partner smoking in pregnancy† | p=0.0794 | p=0.8248 | p=0.0816 | p=0.0242‡ | p=0.0004** | p=0.0049** | p=0.0099** |
| No | 25.1 (5.1) | 7.9 (4.3) | 0.42 (0.11) | 79.5 (11.3) | 0.47 (0.06) | 16.0 (2.5) | 167.8 (9.4) |
| Yes | 25.4 (5.6) | 7.9 (4.7) | 0.42 (0.12) | 80.5 (12.1) | 0.48 (0.07) | 16.3 (2.6) | 167.0 (9.6) |

*p<0.05; **p<0.01; ****p<0.0001.
†Student's t-test
‡Analysis of variance.

whose mother stopped smoking in the pregnancy or smoked from 1 to 14 cigarettes/day (table 3). Although not be objective of the study, we also tested the association between maternal smoking in gestation and overweight in the offspring and found a positive association even after adjusting for confounding factors (OR adjusted: 1.41; 95% CI: 1.26 to 1.57).

Table 4 shows that maternal smoking was associated with most of the body composition outcomes, even after controlling for partner smoking. Moreover, in both cohorts, the magnitude of the associations was stronger for maternal than partner smoking. Because information on partner smoking during pregnancy was not available for the 1982 cohort, we used the information on partner smoking collected in the follow-up visit at 4 years of age.

Concerning the mediation analysis, birth weight captured a small portion of the associations, whereas weight gain in the first 2 years captured most of the association of maternal smoking with BMI (96.2%), fat mass index (71.7%) and waist circumference (86.1) (table 5).

## DISCUSSION

Our findings suggest that exposure to maternal tobacco smoking in utero increases adiposity in early adulthood. These associations were observed even after controlling for several confounding variables. Furthermore, we observed that the magnitude of the association was higher for maternal than paternal smoking and even after controlling for paternal smoking, maternal smoking during pregnancy was associated with body composition in adulthood. Suggesting, therefore, that residual confounding is an unlikely explanation for the association between maternal smoking and offspring body composition.

The results of our study corroborate the findings from previous studies that have observed a positive association between maternal smoking during pregnancy and offspring adiposity.[11 12 16] In relation to central adiposity, it has also been reported higher means waist circumference[16] and android to gynoid fat ratio.[17] Such measures are indicators of fat accumulation in the abdominal region and are more related to cardiovascular risk.[18]

Concerning the mechanisms underlying the association between maternal smoking during pregnancy and offspring adiposity, it has been suggested that this association is due to the nicotine, present in tobacco. In both humans and animals, nicotine when crossing the placenta acts centrally and peripherally as a suppressant of appetite and body weight, resulting in hyperphagia and weight gain when the offspring is no longer exposed to nicotine in postnatal period.[19 20] In addition, exposure to nicotine in gestation may result in increase in body adiposity through alterations in endocrine control of body weight homeostasis.[21] In animals, prenatal exposure to nicotine decreased responsiveness to adrenergic stimuli and promoted rapid weight gain.[22] Similarly, prenatal exposure to nicotine in humans may decrease responsiveness to adrenergic stimuli by epinephrine and norepinephrine, which modulate the mobilisation of lipids from adipose tissue.[23]

It has been reported that low birth weight is associated with increased risk of obesity later in life,[24 25] but in the present study we did not find any evidence that birth weight mediated the observed associations. On the other hand, weight gain in early childhood captured an important proportion of the association of maternal smoking in pregnancy with BMI, fat mass index and waist circumference. Maternal smoking tends to decrease foetal growth, which would be compensated by rapid postnatal weight gain.[26] Because offspring of smoking mothers were shorter, it seems that weight gain was higher than linear growth. Indeed, Adair et al[27] reported that faster relative weight gain in the first 2 years was associated with a higher risk of overweight/obesity in adulthood.

In our study, we observed that lean mass was higher among offspring of mothers who smoked in pregnancy. Leary et al[11] also observed that lean mass was higher among subjects whose mother smoked in the pregnancy. These authors suggested that this association with lean mass would be simply due to the association with total body mass. Indeed, subjects with higher total body mass would have higher fat and lean mass. Concerning the association with height, a negative association has already been reported by previous studies.[12 28]

Our study has many strengths, the data were collected prospectively and information on maternal smoking during pregnancy was collected soon after delivery, with a short recall time. Previous studies on this subject are mainly retrospective, or cross-sectional,[28 29] and data on maternal smoking were gathered with a long recall time, which can lead to misclassification. Second, the high follow-up rates, even after 30 years and the lack of differences in the follow-up rate according to maternal smoking during pregnancy suggests that selection bias is unlikely.[14] Finally, most of the previous studies have evaluated only BMI as a measure of adiposity. We used several anthropometric and body composition measurements, including fat and lean mass measured using DXA. By comparing the estimates of maternal smoking to partner smoking, we increased causal inference and the analysis suggested that the observed associations were unlikely to be due to residual confounding.

With respect to the limitations, information on parental smoking was gathered using a questionnaire and the information was not validated with biochemical markers. Because smoking in pregnancy is a socially disapproved act, some mothers may have occulted the information on prenatal smoking,[30] underestimating the prevalence of smoking during pregnancy. Because this classification error was independent of body composition, which was measured many years later, the misclassification would be non-differential. Because non-differential misclassification tends to underestimate the magnitude of the associations, the observed associations are not due to such error. Another limitation is the fact that information on

**Table 3** Adjusted regression coefficients for anthropometric and body composition measures in adulthood according to maternal smoking during pregnancy

| | Adjusted regression coefficient (95% CI)* | | | | | | |
|---|---|---|---|---|---|---|---|
| | Body mass index (kg/m²) | Fat mass index (kg/m²) | Android to gynoid fat ratio | Waist circumference (cm) | Waist to height ratio | Lean mass index (kg/m²) | Height (cm) |
| Maternal smoking in pregnancy | | | | | | | |
| No | Reference | Reference | Reference | Reference | Reference | Reference | Reference |
| Yes | 0.84 (0.55 to 1.12) | 0.44 (0.23 to 0.64) | 0.016 (0.010 to 0.023) | 1.74 (1.15 to 2.33) | 0.013 (0.010 to 0.017) | 0.33 (0.24 to 0.42) | −0.95 (−1.26 to −0.65) |
| Duration of maternal smoking | | | | | | | |
| Non-smokers | Reference | Reference | Reference | Reference | Reference | Reference | Reference |
| Smoked in part of pregnancy | 1.17 (0.62 to 1.71) | 0.70 (0.30 to 1.11) | 0.013 (0.001 to 0.026) | 2.49 (1.32 to 3.65) | 0.016 (0.009 to 0.023) | 0.34 (0.16 to 0.53) | −0.43 (−1.03 to 0.18) |
| Smoked in whole pregnancy | 0.76 (0.46 to 1.06) | 0.37 (0.15 to 0.59) | 0.017 (0.010 to 0.024) | 1.56 (0.93 to 2.19) | 0.013 (0.009 to 0.016) | 0.32 (0.22 to 0.42) | −1.07 (−1.40 to −0.74) |
| Cigarettes smoked per day | | | | | | | |
| None | Reference | Reference | Reference | Reference | Reference | Reference | Reference |
| 1–14 cigarettes/day | 0.89 (0.60 to 1.19) | 0.46 (0.24 to 0.69) | 0.018 (0.011 to 0.025) | 1.86 (1.23 to 2.50) | 0.014 (0.010 to 0.017) | 0.34 (0.24 to 0.44) | −0.81 (−1.15 to −0.49) |
| ≥15 cigarettes/day | 0.62 (0.09 to 1.15) | 0.32 (−0.08 to 0.71) | 0.010 (−0.003 to 0.022) | 1.24 (0.12 to 2.36) | 0.012 (0.005 to 0.019) | 0.30 (0.12 to 0.47) | −1.52 (−2.11 to −0.94) |

*Adjusted for sex, age (Cohort), family income at birth, maternal skin colour, maternal schooling, maternal age at childbirth, parity, pre-gestational nutritional status and maternal height.

**Table 4** Adjusted regression coefficients for anthropometric and body composition measures in adulthood according to maternal smoking during pregnancy and partner smoking

| | Adjusted regression coefficient (95% CI) | | | | | | |
| --- | --- | --- | --- | --- | --- | --- | --- |
| | Body mass index (kg/m²) | Fat mass index (kg/m²) | Android to gynoid fat ratio | Waist circumference (cm) | Waist to height ratio | Lean mass index (kg/m²) | Height (cm) |
| **1982 cohort** | | | | | | | |
| Maternal smoking in pregnancy* | | | | | | | |
| No | Reference | Reference | Reference | Reference | Reference | Reference | Reference |
| Yes | 0.65 (0.24 to 1.06) | 0.30 (0.01 to 0.59) | 0.020 (0.011 to 0.030) | 1.54 (0.67 to 2.41) | 0.012 (0.006 to 0.017) | 0.27 (0.13 to 0.41) | −0.75 (−1.17 to −0.33) |
| Partner smoking at 4 years* | | | | | | | |
| No | Reference | Reference | Reference | Reference | Reference | Reference | Reference |
| Yes | −0.04 (−0.47 to 0.39) | 0.01 (−0.29 to 0.33) | 0.001 (−0.010 to 0.011) | −0.05 (−0.97 to 0.88) | 0.001 (−0.005 to 0.006) | 0.02 (−0.12 to 0.17) | −0.11 (−0.56 to 0.34) |
| Maternal smoking in pregnancy† | | | | | | | |
| No | Reference | Reference | Reference | Reference | Reference | Reference | Reference |
| Yes | 0.64 (0.19 to 1.10) | 0.24 (−0.09 to 0.56) | 0.016 (0.005 to 0.027) | 1.60 (0.62 to 2.57) | 0.012 (0.006 to 0.018) | 0.24 (0.09 to 0.39) | −0.73 (−1.21 to −0.26) |
| Partner smoking at 4 years‡ | | | | | | | |
| No | Reference | Reference | Reference | Reference | Reference | Reference | Reference |
| Yes | −0.04 (−0.47 to 0.39) | 0.02 (−0.29 to 0.33) | 0.001 (−0.010 to 0.011) | −0.05 (−0.97 to 0.87) | 0.001 (−0.005 to 0.006) | 0.02 (−0.12 to 0.17) | −0.11 (−0.55 to 0.35) |
| **1993 cohort** | | | | | | | |
| Maternal smoking in pregnancy* | | | | | | | |
| No | Reference | Reference | Reference | Reference | Reference | Reference | Reference |
| Yes | 1.04 (0.66 to 1.42) | 0.38 (0.25 to 0.51) | 0.014 (0.006 to 0.023) | 1.99 (1.18 to 2.80) | 0.016 (0.011 to 0.021) | 0.38 (0.25 to 0.51) | −1.21 (−1.66 to −0.76) |
| Partner smoking in pregnancy* | | | | | | | |
| No | Reference | Reference | Reference | Reference | Reference | Reference | Reference |
| Yes | 0.48 (0.11 to 0.85) | 0.20 (−0.09 to 0.48) | 0.006 (−0.002 to 0.014) | 1.23 (0.44 to 2.01) | 0.009 (0.004 to 0.013) | 0.21 (0.09 to 0.34) | −0.28 (−0.71 to 0.15) |
| Maternal smoking in pregnancy§ | | | | | | | |
| No | Reference | Reference | Reference | Reference | Reference | Reference | Reference |
| Yes | 1.01 (0.61 to 1.42) | 0.36 (0.23 to 0.50) | 0.015 (0.006 to 0.024) | 1.92 (1.05 to 2.78) | 0.015 (0.009 to 0.020) | 0.36 (0.23 to 0.50) | −1.13 (−1.61 to −0.65) |
| Partner smoking in pregnancy‡ | | | | | | | |
| No | Reference | Reference | Reference | Reference | Reference | Reference | Reference |
| Yes | 0.28 (0.09 to 0.67) | 0.07 (−0.22 to 0.36) | 0.003 (−0.005 to 0.012) | 0.85 (0.06 to 1.66) | 0.006 (0.001 to 0.011) | 0.14 (0.02 to 0.27) | −0.06 (−0.50 to 0.38) |

*Adjusted for sex, family income at birth, maternal skin colour, maternal schooling, maternal age at childbirth, parity, pre-gestational nutritional status and maternal height.
†Adjusted for sex, family income at birth, maternal skin colour, maternal schooling, maternal age at childbirth, parity, pre-gestational nutritional status, maternal height and partner smoking at 4 years.
‡Adjusted for sex, family income at birth, maternal skin colour, maternal schooling, maternal age at childbirth, parity, pre-gestational nutritional status, maternal height and maternal smoking in pregnancy.
§Adjusted for sex, family income at birth, maternal skin colour, maternal schooling, maternal age at childbirth, parity, pre-gestational nutritional status, maternal height and partner smoking in pregnancy.

**Table 5** Analysis of mediation in the association between maternal smoking during pregnancy and body composition measures at adulthood in the 1982 (at 30 years) and 1993 (22 years) Pelotas birth cohorts, Brazil

| Anthropometric and body composition measures | Mediators | G-computation estimate (95% CI) | | Mediated effect (%) |
|---|---|---|---|---|
| | | Natural direct effect | Natural indirect effect | |
| Body mass index (kg/m$^2$) | Birth weight | 0.79 (0.44 to 1.15) | 0.13 (−0.05 to 0.31) | 14.1 |
| | Weight gain from birth to 2 years | 0.01 (−0.53 to 0.54) | 0.21 (−0.08 to 0.50) | 96.2 |
| Fat mass index (kg/m$^2$) | Birth weight | 0.44 (0.19 to 0.70) | −0.03 (−0.16 to 0.10) | 7.7 |
| | Weight gain from birth to 2 years | −0.02 (−0.40 to 0.35) | −0.06 (−0.27 to 0.14) | 71.7 |
| Android to gynoid fat ratio | Birth weight | 0.014 (0.006 to 0.022) | −0.002 (−0.006 to 0.002) | 17.1 |
| | Weight gain from birth to 2 years | 0.018 (0.005 to 0.030) | 0.001 (−0.006 to 0.007) | 3.4 |
| Waist circumference (cm) | Birth weight | 2.02 (1.28 to 2.76) | 0.18 (−0.22 to 0.57) | 8.1 |
| | Weight gain from birth to 2 years | 0.12 (−1.03 to 1.26) | 0.72 (0.09 to 1.34) | 86.1 |
| Waist to height ratio | Birth weight | 0.012 (0.007 to 0.016) | 0.001 (−0.002 to 0.003) | 3.0 |
| | Weight gain from birth to 2 years | 0.002 (−0.005 to 0.009) | 0.003 (−0.001 to 0.007) | 56.0 |
| Lean mass index (kg/m$^2$) | Birth weight | 0.37 (0.25 to 0.48) | 0.04 (−0.02 to 0.10) | 8.9 |
| | Weight gain from birth to 2 years | 0.16 (−0.02 to 0.34) | −0.02 (−0.11 to 0.08) | 12.7 |
| Height (cm) | Birth weight | −0.40 (−0.78 to −0.02) | −0.18 (−0.39 to 0.03) | 30.9 |
| | Weight gain from birth to 2 years | −0.93 (−1.52 to −0.33) | 0.10 (−0.24 to 0.43) | 11.5 |

partner smoking was obtained at different times in the two cohorts. However, it is likely that parents who smoked during pregnancy remained smokers after birth and the first years of the offspring's life. In addition, no discrepant results were observed regarding the effect of paternal smoking among the two cohorts.

In conclusion, we observed an association between maternal smoking in pregnancy and body composition measures in adulthood. The specificity of the association for maternal, in relation to paternal smoking, suggested that this association is not due to residual confounding. Analysis of mediation suggests the weight gain from birth to 2 years is an important mediator of the association between maternal smoking during pregnancy and adiposity in adulthood.

**Acknowledgements** We acknowledge the contributions of 1982 and 1993 Pelotas Cohorts participants, other researchers and staff.

**Contributors** EISM designed the study, performed the statistical analysis, interpretation of the results and drafted the manuscript. BLH and NPL designed the study, helped the data analysis and participated in the preparation of the manuscript. BLH coordinated the follow-up of the 1982 cohort and AMBM coordinated the follow-up of the 1993 cohort. AMBM, HDG, FCW and MCA helped in the data acquisition and interpretation of the data. All authors revised and approved the final version of the manuscript. Each author contributed important intellectual content during manuscript drafting or revision and accepts accountability for the overall work by ensuring that questions pertaining to the accuracy or integrity of any portion of the work are appropriately investigated and resolved.

**Funding** This article is based on data from the study 'Pelotas Birth Cohort, 1982 and 1993' conducted by the Postgraduate Program in Epidemiology at Federal University of Pelotas with the collaboration of the Brazilian Public Health Association (ABRASCO). From 2004 to 2016, the Wellcome Trust (086974/Z/08/Z) supported the Pelotas birth cohort study. The International Development Research Center, World Health Organization, Overseas Development Administration, European Union, National Support Program for Centers of Excellence (PRONEX), the Brazilian National Research Council (CNPq) and the Brazilian Ministry of Health supported previous phases of the study. This study was financed in part by theCoordenação de Aperfeiçoamento de Pessoal de Nível Superior - Brasil (CAPES) -Finance Code 001.

**Competing interests** None declared.

**Patient consent for publication** Obtained.

**Provenance and peer review** Not commissioned; externally peer reviewed.

**Data sharing statement** Data are available upon reasonable request.

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
