## [Reviewer comments · BMJ Open]

ARTICLE DETAILS

TITLE (PROVISIONAL)	Maternal smoking during pregnancy and offspring body composition in adulthood: Results from two birth cohort studies
AUTHORS	Magalhães, Elma Izze; Lima, Natália; Menezes, Ana Maria; Goncalves, Helen; Wehrmeister, Fernando; Assuncao, M.C; Horta, Bernardo

VERSION 1 – REVIEW

REVIEWER	Thorhallur Halldorsson University of Iceland
REVIEW RETURNED	25-May-2018

GENERAL COMMENTS	This is an interesting paper describing the association between maternal smoking in pregnancy and offspring anthropometry at adult age. Although the study has many strengths in terms of long term follow-up there is some room for improvement and clarification of their findings. Firstly, the data collections need to be more clearly described in terms of how many participants were invited, how many participated and how many dropped out during follow-up. This information is scattered through various sections of the paper. In addition it is important to know if there were any major difference between those initially recruited into the study and those mothers whose offspring did not participate at follow-up As confounding by lifestyle and income is important to address (rightly acknowledged in the introduction) it is also important to present the data in a way that the likelihood of such confounding (including residual confounding) can be evaluated independently by the reader. This means showing how income, education and partner smoking are distributed among mothers who did not smoke, partly smoked and smoked daily during pregnancy. In the 1993 cohort partner smoking was assessed in pregnancy and was also strongly related to offspring anthropometry at adult age. Why are those results not discussed? Firstly, these results do suggest a non-causal relationship between partner smoking and offspring weight. Secondly it is surprising that the adjustment for partner smoking was not influential. Can this be discussed in more detail. The lack of association for partner smoking assessed
---

at 4 years in the other cohort does indeed point towards some sort of confounding

To convince the readers that confounding is not an issue it would also be relevant to presented analyses stratified by partner smoking and familial income. That is, Is the association between maternal smoking and offspring later anthropometry comparable depending on if the partner smoked or not in pregnancy; and stable across familial income (this does perhaps not address confounding directly but is important in terms to evaluate the stability of the associations reported).

I am also a bit surprised why only effect estimates for continuous outcomes are reported. What does a mean difference of around 1 bmi unit mean in terms of risk of obesity in adult life? Such presentation of results is more meaningful in terms of public health implications.

Other comments

Abstract:

When describing participants. Where the offspring exactly 30 and 22 years of age at follow-up. If not be more specific when describing the follow-up (at what year and what was the range with respect to time, did all the births only occur in 1982 and 1993?)

In the abstract results it is quite difficult to but the estimates as there are no units given /kg/m² for BMI and so on). In addition, we don't know anything about the mean measures (for BMI and the other outcomes) so it is difficult to but the effect estimates in any context.

Introduction

Methods

The description of the study design and population is too short. Since information on weight gain for the first 2 years is being reported and weight at adult age we need more detailed information on the follow-up (what was measured and how in terms of clinical versus self-report). We also need more detailed information on information on social characteristics that were recorded, which is essential in terms of tackling confounding (which the introduction addresses nicely).

Also, when describing the study population, you need to explain how many were invited to participate originally, how many accepted the invitation and then explain the dropout rate in the later follow-up. This should be done in the method section when describing the cohort but not in the results section (as It is now).

In the section "exposure variables" it would be relevant to emphasise that information on maternal smoking was assessed shortly after pregnancy (retrospectively according to the study design section)

When was the information on partner smoking collected in the 1993 study and was it based on the mothers or the partners own report?

	On confounder control. Why were partners smoking habits not considered as confounder in your analyses? Statistical analyses. When you say that the association for smoking was not modified by the cohort it might be clearer to say that “both cohorts showed similar association between maternal smoking and offspring anthropometry”. Statistical analyses: What is a post-confounder and how are such confounders different from those assessed in pregnancy? Results and Discussion Well written and mostly clear no comments. Comments on the tables. What is the first table (prior to table 1)? Why is information on offspring sex not just presented in Table 1? In table 1 you need to indicate that the numbers are n (%) for the categorical variables and mean(SD) for the continuous ones. Unclear now as presented. Why is information on important confounders such as income and education not presented in this table Table 2: What does the p-values that is under Total stand for? Please add information on partner smoking to table 2 and 3 (showing how it is related to offspring anthropometric measures)
--	--

REVIEWER	Michael Chaiton University of Toronto, Canada
REVIEW RETURNED	31-May-2018

GENERAL COMMENTS	The article, “Maternal smoking during pregnancy and offspring body composition in adulthood: Results from two birth cohort studies” adds significantly to the literature on the relationship of maternal smoking during pregnancy and offspring obesity. Literature review could be clarified. Individual studies are compared with meta-analysis which overstates the degree of disagreement in the literature. The sentence on line 187 p 8 is confusing: “Moreover, 32.4% of the partners smoked during pregnancy in the 1993 cohort and 58.8% of the partners were smoker at the follow-up visit at 4 years of age”. The lack of data on partner smoking at birth in 1982 should be described in the variables section. However, why the big jump in smoking prevalence among this group over a four year period? Initiation among this group seem unlikely. Description of the procedure for multiple regression and g computation were very brief and should be more complete. In particular it would be helpful to introduce g computation, describe which package was used as it is not part of the standard Stata 14 set, and justify why g computation compared to a semiparametric method with less assumptions such as IPW of marginal structural models. The authors could also consider a figure to show the timing of pre and post confounding.
---

	One additional limitation is that exposure to secondhand smoke through paternal smoking could have an effect, so it should be noted that comparisons of maternal to paternal smoking would underestimate the effect. Similarly, the analysis of duration is limited because the window of exposure is unknown. It is likely that only part of the period of pregnancy is at risk for prenatal exposure, and so even smoking for a portion of the pregnancy may mean smoking for the entire risk period. As the authors highlight the lost to follow up rate as a strength of the study, it is important to include in the text additional analysis of the characteristics of those lost and not lost to follow up. Although it's not stated in the inclusion criteria, it seems that another selection bias issue that should be mentioned is that the study only includes those who survived to term? Additional detail from the mediation analysis could be included in the body text, particularly the % of the associations captured by BMI, fat mass index, and waist circumference. The discussion on the finding on lean fat mass was confusing and the implications are difficult to interpret—is there an association with total body mass? That seems hard to square with the findings of a negative association with height. P. 11 line 256. While some of the literature deals with cross sectional and retrospective studies, that does not represent a significant portion of the literature reviewed in the meta-analyses on longitudinal studies of this association.
--	---

REVIEWER	Ester di Giacomo University of Milano Bicocca, Italy
REVIEW RETURNED	09-Jun-2018

GENERAL COMMENTS	I read the work with interest. It is well planned and carried out. Results are appealing. I just have two small suggestions to report: Firstly, in methods you have to report all the details without referencing to other studies. Then, the statistical values (eg p values) of the strongest results should be reported in the text (ref line 194/196)
---

VERSION 1 – AUTHOR RESPONSE

Reviewer: 1
Reviewer Name: Thorhallur Halldorsson
Institution and Country: University of Iceland

Comments and answers to the reviewer:

1. The data collections need to be more clearly described in terms of how many participants were invited, how many participated and how many dropped out during follow-up. This information is scattered through various sections of the paper. In addition is important to know if there were any major difference between those initially recruited into the study and those mothers who's offspring did not participate at follow-up.

As suggested by the reviewer, in the methods section we provided information on how many participants were invited, how many participated and how many dropped out during follow-up. Furthermore, the Supplementary Table 1 reported the follow-up rate at each visit of the 1982 and 1993 cohorts.

We also included the Supplementary Table 2 that reports the follow-up rate in the last visit of each cohort, according to baseline characteristics.

2. As confounding by lifestyle and income is important to address (rightly acknowledged in the introduction) it is also important to present the data in a way that the likelihood of such confounding (including residual confounding) can be evaluated independently by the reader. This means showing how income, education and partner smoking are distributed among mothers who did not smoked, partly smoked and smoked daily during pregnancy.

As suggested by the reviewer, we added a table (Supplementary Table 3) reporting on the association between maternal smoking during pregnancy and maternal socioeconomic characteristics.

3. In the 1993 cohort partners smoking was assessed in pregnancy and was also strongly related to offspring anthropometry at adult age. Why are those results not discussed? Firstly, these results do suggest a non-causal relationship between partner smoking and offspring weight. Secondly it is surprising that the adjustment for partners smoking was not influential. Can this be discussed in more detail. The lack of association for partner smoking assessed at 4 years in the other cohort does indeed point towards some sort of confounding.

Paternal smoking was associated with offspring anthropometry, but after adjusting for maternal smoking the magnitude of the association reduced, whereas for maternal smoking the estimates slightly changed. These findings suggests that part of the association between paternal smoking and offspring anthropometry as due to the association of paternal with maternal smoking.

4. To convince the readers that confounding is not an issue it would also be relevant to presented analyses stratified by partner smoking and familial income. That is, Is the association between maternal smoking and offspring later anthropometry comparable depending on if the partner smoked or not in pregnancy; and stable across familial income (this does perhaps not address confounding directly but is important in terms to evaluate the stability of the associations reported).

As requested by the reviewer, we added tables below (Table 1 and Table 2 attached in this response) reporting the results of analyses stratified by partner smoking and familial income tertile.

Table 1. Adjusted regression coefficients for anthropometric and body composition measures in adulthood according to maternal smoking during pregnancy and stratified by partner smoking and familial income tertile in the 1982 cohort.

Adjusted regression coefficient (95% confidence interval) ^a							
	Body mass index (kg/m ²)	Fat mass index (kg/m ²)	Android to gynoid fat ratio	Waist circumference (cm)	Waist to height ratio	Lean mass index (kg/m ²)	Height (cm)
Partner smoker at 4 years and familiar income 1 st tertile							
Maternal smoking in pregnancy							
No	Reference	Reference	Reference	Reference	Reference	Reference	Reference
Yes	0.24 (-0.94 ; 1.41)	-0.40 (-1.25 ; 0.45)	-0.037 (-0.031 ; 0.024)	0.92 (-1.49 ; 3.34)	0.006 (-0.009 ; 0.021)	0.09 (-0.31 ; 0.49)	-0.26 (-1.59 ; 1.07)
Partner smoker at 4 years and familiar income 2 nd tertile							
Maternal smoking in pregnancy							
No	Reference	Reference	Reference	Reference	Reference	Reference	Reference
Yes	0.86 (-0.12 ; 1.83)	0.25 (-0.45 ; 0.94)	-0.001 (-0.024 ; 0.022)	1.98 (-0.19 ; 4.15)	0.014 (0.001 ; 0.027)	0.29 (-0.03 ; 0.62)	-0.65 (-1.75 ; 0.46)

Partner smoker at 4 years and familiar income 3rd tertile
Maternal smoking in pregnancy

No	Reference	Reference	Reference	Reference	Reference	Reference	Reference
Yes	0.45 (-0.50 ; 1.39)	0.41 (-0.31 ; 1.12)	0.028 (0.004 ; 0.052)	1.14 (-0.88 ; 3.16)	0.010 (-0.003; 0.022)	0.08 (-0.26 ; 0.42)	-0.93 (- 2.21 ; 0.34)

Partner non- smoker at 4 years and familiar income 1st tertile
Maternal smoking in pregnancy

No	Reference	Reference	Reference	Reference	Reference	Reference	Reference
Yes	1.48 (0.15 ; 2.81)	1.02 (0.02 ; 2.01)	0.047 (0.015 ; 0.079)	3.89 (1.10 ; 6.69)	0.026 (0.009; 0.044)	0.47 (0.02 ; 0.91)	-0.12 (- 1.56 ; 1.33)

Partner non-smoker at 4 years and familiar income 2nd tertile
Maternal smoking in pregnancy

No	Reference	Reference	Reference	Reference	Reference	Reference	Reference
Yes	0.55 (-0.59 ; 1.69)	0.17 (-0.63 ; 0.97)	0.019 (-0.008 ; 0.046)	1.52 (-1.02 ; 4.06)	0.011 (-0.003; 0.026)	0.44 (0.05 ; 0.82)	-0.57 (- 1.94 ; 0.79)

Partner non-smoker at 4 years and familiar income 3rd tertile
Maternal smoking in pregnancy

No	Reference	Reference	Reference	Reference	Reference	Reference	Reference
Yes	0.54 (-0.83 ; 1.90)	0.19 (-0.80 ; 1.19)	0.023 (-0.009 ; 0.054)	0.53 (-2.48 ; 3.54)	0.006 (-0.012; 0.023)	0.18 (-0.22 ; 0.59)	-0.92 (- 2.48 ; 0.64)

^a Adjusted for: Sex, maternal skin colour, maternal schooling, maternal age at childbirth, parity, pre-gestational nutritional status and maternal height.

Table 2. Adjusted regression coefficients for anthropometric and body composition measures in adulthood according to maternal smoking during pregnancy and stratified by partner smoking and familial income tertile in the 1993 cohort.

	Adjusted regression coefficient (95% confidence interval) ^a						
	Body mass index (kg/m ²)	Fat mass index (kg/m ²)	Android to gynoid fat ratio	Waist circumference (cm)	Waist to height ratio	Lean mass index (kg/m ²)	Height (cm)
Partner smoker in pregnancy and familiar income 1 st tertile							
Maternal smoking in pregnancy							
No	Reference	Reference	Reference	Reference	Reference	Reference	Reference
Yes	1.47 (0.69 ; 2.25)	0.97 (0.38 ; 1.56)	0.018 (0.001 ; 0.035)	2.50 (0.85 ; 4.15)	0.018 (0.008 ; 0.028)	0.47 (0.21 ; 0.72)	-0.96 (-1.95 ; 0.03)
Partner smoker in pregnancy and familiar income 2 nd tertile							
Maternal smoking in pregnancy							
No	Reference	Reference	Reference	Reference	Reference	Reference	Reference
Yes	1.33 (0.26 ; 2.41)	0.35 (-0.46 ; 1.16)	0.011 (-0.011 ; 0.034)	2.19 (-0.03 ; 4.42)	0.019 (0.005 ; 0.032)	0.45 (0.10 ; 0.80)	-1.87 (-3.14 ; -0.60)
Partner smoker in pregnancy and familiar income 3 rd tertile							
Maternal smoking in pregnancy							
No	Reference	Reference	Reference	Reference	Reference	Reference	Reference
Yes	0.46 (-0.68 ; 1.60)	0.38 (-0.48 ; 1.24)	0.012 (-0.015 ; 0.039)	0.47 (-2.10 ; 3.04)	0.009 (-0.007 ; 0.025)	0.35 (-0.05 ; 0.74)	-2.62 (-4.09 ; 1.14)
Partner non-smoker in pregnancy and familiar income 1 st tertile							

Maternal smoking in pregnancy

No	Reference	Reference	Reference	Reference	Reference	Reference	Reference
Yes	1.07 (0.02 ; 2.12)	0.78 (-0.03 ; 1.59)	0.023 (-0.001 ; 0.046)	2.37 (0.15 ; 4.59)	0.018 (0.004 ; 0.031)	0.39 (0.05 ; 0.73)	-1.13 (-2.43 ; 0.18)

Partner non-smoker in pregnancy and familiar income 2nd tertile

Maternal smoking in pregnancy

No	Reference	Reference	Reference	Reference	Reference	Reference	Reference
Yes	0.17 (-1.00 ; 1.33)	0.23 (-0.67 ; 1.12)	-0.001 (-0.026 ; 0.025)	0.71 (-1.76 ; 3.18)	0.003 (-0.012 ; 0.018)	0.26 (-0.14 ; 0.65)	0.38 (-1.17 ; 1.92)

Partner non-smoker in pregnancy and familiar income 3rd tertile

Maternal smoking in pregnancy

No	Reference	Reference	Reference	Reference	Reference	Reference	Reference
Yes	0.89 (-0.13 ; 1.91)	0.82 (-0.01 ; 1.65)	0.032 (0.009 ; 0.056)	1.81 (-0.36 ; 3.99)	0.016 (0.003 ; 0.029)	0.16 (-0.21 ; 0.53)	-1.99 (-3.52 ; 0.47)

^a Adjusted for: Sex, maternal skin colour, maternal schooling, maternal age at childbirth, parity, pre-gestational nutritional status and maternal height.

5. I am also a bit surprised why only effect estimates for continuous outcomes are reported. What does a mean difference of around 1 bmi unit mean in terms of risk of obesity in adult life? Such presentation of results is more meaningful in terms of public health implications.

We also tested the association of maternal smoking in gestation and overweight of offspring, which was also significant (OR Adjusted: 1.41; 95%CI: 1.26 to 1.57), but we chose to report the effect of exposure to pre-natal maternal smoking on measures of continuous body composition in view of the lacuna in the literature for these findings. However, although it is not the purpose of the study, we report this result in the article to provide this additional information to the reader.

Other comments and respective answers:

Abstract

1. When describing participants. Where the offspring exactly 30 and 22 years of age at follow-up. If not be more specific when describing the follow-up (at what year and what was the range with respect to time, did all the births only occur in 1982 and 1993?)

As mentioned in the methods section, only those births occurring in 1982 and 1993 were included in the study. With respect to the exact age at follow, as requested by the reviewer we included the exact information (At a mean age of 30.2 and 22.6 years, the 1982 and 1993 cohorts, respectively) in the abstract.

2. In the abstract results it is quite difficult to but the estimates as there are no units given /kg/m² for BMI and so on). In addition, we don't know anything about the mean measures (for BMI and the other outcomes) so it is difficult to but the effect estimates in any context.

As requested by the reviewer, we added the units of measure in the estimates of effect of the body composition measurements in the abstract, except for android to gynoid fat ratio and waist to height ratio), which express the ratio of two measurements and have no unit of measure.

Methods

1. The description of the study design and population is too short. Since information on weight gain for the first 2 years is being reported and weight at adult age we need more detailed information on the follow-up (what was measured and how in terms of clinical versus self-report). We also need more detailed information on information on social characteristics that were recorded, which is essential in terms of tackling confounding (which the introduction addresses nicely).

As suggested by the reviewer, we added requested information.

2. Also, when describing the study population, you need to explain how many were invited to participate originally, how many accepted the invitation and then explain the dropout rate in the later follow-up. This should be done in the method section when describing the cohort but not in the results section (as It is now).

As mentioned previously, in the methods section we added information on how many participants were invited, how many participated and how many dropped out during follow-up.

3. In the section “exposure variables” it would be relevant to emphasise that information on maternal smoking was assessed shortly after pregnancy (retrospectively according to the study design section).

Conforming solicited, we emphasized this information.

4. When was the information on partner smoking collected in the 1993 study and was it based on the mothers or the partners own report?

In 1993, the mother provided information on paternal smoking during the interview in the perinatal study. This detail was clarified in the text.

5. On confounder control. Why were partners smoking habits not considered as confounder in your analyses?

We considered partner smoking as a negative confounding control, and presented the results in Table 4.

6. Statistical analyses: When you say that the association for smoking was not modified by the cohort it might be clearer to say that “both cohorts showed similar association between maternal smoking and offspring anthropometry”.

As suggested by the reviewer, we changed the sentence.

7. Statistical analyses: What is a post-confounder and how are such confounders different from those assessed in pregnancy?

In mediation analysis, post-confounders are those variables that confound the relationship between the mediator and the outcome. This clarification was added in the text.

Comments on the tables

1. What is the first table (prior to table 1)? Why is information on offspring sex not just presented in Table 1?

Table 1 show the characteristics of population included in the study. Information on sex was presented in this table.

2. In table 1 you need to indicate that the numbers are n (%) for the categorical variables and mean(SD) for the continuous ones. Unclear now as presented.

Conforming solicited by reviewer, this information was added.

3. Why is information on important confounders such as income and education not presented in this table

As suggested by the reviewer, we added this information in Table 1.

4. Table 2: What does the p-values that is under Total stand for?

This P-value refer to association between each exposure variable and offspring body composition. The line with mean and standard deviation total of each body composition measure was dropped in table for not confound reader.

5. Please add information on partner smoking to table 2 and 3 (showing how it is related to offspring anthropometric measures).

As requested by reviewer, partner smoking data were added in Table 2. The adjusted estimates were not added in Table 3, as they are already presented in Table 4.

Reviewer: 2

Reviewer Name: Michael Chaiton

Institution and Country: University of Toronto, Canada

Comments and answers to the reviewer:

1. Literature review could be clarified. Individual studies are compared with meta-analysis which overstates the degree of disagreement in the literature.

We did not compare meta-analyses and individual studies, we just reported on the findings of these studies and highlighted the relevance of dealing of residual confounding in the interpretation of the findings on this association.

2. The sentence on line 187 p 8 is confusing: "Moreover, 32.4% of the partners smoked during pregnancy in the 1993 cohort and 58.8% of the partners were smoker at the follow-up visit at 4 years of age".

This sentence was revised.

3. The lack of data on partner smoking at birth in 1982 should be described in the variables section. However, why the big jump in smoking prevalence among this group over a four year period? Initiation among this group seem unlikely.

We agree with the reviewer comment that smoking initiation at this age is unlikely. A possible explanation for the difference in the prevalence of partner smoking between 1982 and 1993 was the sharp decrease in the prevalence of smoking among males that was occurring in Brazil.

4. Description of the procedure for multiple regression and g computation were very brief and should be more complete. In particular it would be helpful to introduce g computation, describe which package was used as it is not part of the standard Stata 14 set, and justify why g computation compared to a semiparametric method with less assumptions such as IPW of marginal structural models. The authors could also consider a figure to show the timing of pre and post confounding.

As suggested by the reviewer, we added the information on the Stata package that provides the g-computation command. We used this approach because it allows the control for post-confounders.

5. One additional limitation is that exposure to secondhand smoke through paternal smoking could have an effect, so it should be noted that comparisons of maternal to paternal smoking would underestimate the effect. Similarly, the analysis of duration is limited because the window of exposure is unknown. It is likely that only part of the period of pregnancy is at risk for prenatal exposure, and so even smoking for a portion of the pregnancy may mean smoking for the entire risk period.

We agree with the reviewer that secondhand smoking could have an effect. Indeed, even after controlling for maternal smoking, we observed that paternal smoking was slightly with some of the offspring outcomes.

6. As the authors highlight the lost to follow up rate as a strength of the study, it is important to include in the text additional analysis of the characteristics of those lost and not lost to follow up. Although it's not stated in the inclusion criteria, it seems that another selection bias issue that should be mentioned is that the study only includes those who survived to term?

We added information on the losses to follow-up. With respect to selection bias, with the due respect, we do not agree with the reviewer. By studying offspring anthropometry in early adulthood, those who did not survive to that age are naturally excluded.

7. Additional detail from the mediation analysis could be included in the body text, particularly the % of the associations captured by BMI, fat mass index, and waist circumference.

As suggested by the reviewer, these information was added in the text.

8. The discussion on the finding on lean fat mass was confusing and the implications are difficult to interpret—is there an association with total body mass? That seems hard to square with the findings of a negative association with height. P. 11 line 256. While some of the literature deals with cross sectional and retrospective studies, that does not represent a significant portion of the literature reviewed in the meta-analyses on longitudinal studies of this association.

Considering the reviewer's recommendation, this paragraph was revised. Conforming discussed, this association is related to the fact that subjects with higher lean mass also have more total body mass.

Reviewer: 3

Reviewer Name: Ester di Giacomo

Institution and Country: University of Milano Bicocca, Italy

Comments and answers to the reviewer:

1. I just have two small suggestions to report:

1.1 Firstly, in methods you have to report all the details without referencing to other studies.

As suggested by the reviewer, we expanded the methods section.

1.2 Then, the statistical values (eg p values) of the strongest results should be reporter in the text (ref line 194/196)

As solicited this values was added in the text.

VERSION 2 – REVIEW

REVIEWER	Ester di Giacomo PhD program in Neuroscience-University of Milan Bicocca (Italy)
REVIEW RETURNED	03-Aug-2018

GENERAL COMMENTS	The reviewer completed the checklist but made no further comments.
--

REVIEWER	Aaron Wendelboe University of Oklahoma Health Sciences Center USA
REVIEW RETURNED	09-May-2019

GENERAL COMMENTS	The format of the tables is not correct in the PDF I reviewed and made it hard to assess clearly.
---